# Psychosomatic syndromes are associated with IL-6 pro-inflammatory cytokine in heart failure patients

**Mario Altamura**[1]*, **Giovanna D'Andrea**[1], **Eleonora Angelini**[1], **Fabio M. P. Tortorelli**[1], **Angela Balzotti**[1], **Piero Porcelli**[2], **Maurizio Margaglione**[1], **Natale D. Brunetti**[3], **Tommaso Cassano**[3], **Antonello Bellomo**[1]

1 Department of Clinical and Experimental Medicine, University of Foggia, Foggia, Italy, 2 Department of Psychological, Health and Territorial Sciences, D'Annunzio University of Chieti–Pescara, Chieti, Italy, 3 Department of Medical and Surgical Sciences, University of Foggia, Foggia, Italy

* mario.altamura@unifg.it

**Data Availability Statement:** All relevant data are within the paper and its Supporting information files.

**Funding:** The author(s) received no specific funding for this work.

## Abstract

Psychosomatic syndromes have emerged as an important source of comorbidity in cardiac patients and have been associated with increased risk for adverse outcomes in patients with heart failure (HF). Understanding of the mechanisms underlying this connection is limited, however immune activity represents a possible pathway. While there have been numerous studies connecting immune activity to psychosomatic psychopathology, there is a lack of research on patients with HF. We examined forty-one consecutive outpatients affected by HF. We assessed psychosomatic psychopathology using the Diagnostic Criteria for Psychosomatic Research (DCPR) and the Patient Health Questionnaire-15 (PHQ-15). The Psychosocial Index (PSI) was used for assessing stress and psychosocial dimensions. Depression was evaluated with Beck Depression Inventory-II (BDI-II). Circulating levels of proinflammatory cytokines IL-6 and TNF-alpha were ascertained. Univariate and multivariable regression models were used to test for associations between inflammatory cytokines and psychosomatic psychopathology (i.e., DCPR syndromes, PHQ-15) and psychological dimensions (i.e., BDI-II, PSI). A significant positive correlation was found between IL-6 levels and psychosomatic psychopathology even when controlling for any confounding variables (i.e., Body-mass index (BMI), New York Heart Association (NYHA) class, smoking habits, alcohol consumption, statin use, aspirin use, beta blockers use, age, and gender). In contrast, the associations between TNF-alpha levels were non-significant. These findings can contribute to research in support of a psychoneuroimmune connection between psychosomatic psychopathology and HF. Findings also suggest the possibility that elevated IL-6 levels are more relevant for the pathogenesis of psychosomatic syndromes than for depression in patients with HF.

## Introduction

An extensive literature suggests that psychological risk factors (e.g., depression, negative personality traits) play a critical role in the progression and mortality of cardiovascular disease

**Competing interests:** The authors have declared that no competing interests exist.

(CVD) and heart failure (HF) [1, 2]. Evidence suggests that depressive symptoms might be associated with adverse outcomes and disease severity in patients with HF, independently of biomedical risk factors [2–4]. The profile of psychosocial risk factors for CVD revealed relevant variability between different diagnostic groups of cardiovascular diseases. For instance, Stauber et al., 2012 [5] compared psychosocial risk factors for CVD across the affective spectrum (e.g, depression, anxiety, vital exhaustion), personality characteristics (ie, hostility, type D personality), and social support between 3 groups of patients with a primary diagnosis of coronary heart disease (CHD), heart failure and peripheral arterial disease. The authors reported that relative to patients with peripheral arterial disease, those with HF showed greater exhaustion and lower positive affect. This clearly shows the relative importance of psychosomatic risk factors (e.g., vital exhaustion) versus other variables in patients with HF.

A substantial number of clinical studies have clearly established the involvement of cytokines in the pathophysiology of major depression and ischemic CVD with and without comorbid depression [6–8]. Evidence shows that proinflammatory cytokines are important determinants of severity and prognosis of CVD [9–11]. Plasma levels of cytokines are strong predictors of patients' functional class (NYHA) and also predictors of left ventricular function (LVEF) decline [12]. Furthermore, and maybe even more important, a growing body of evidence indicate that the association of psychological risk factors with adverse cardiovascular events is due to immunological processes, specifically inflammation [4, 6, 7, 13–15].

Little is known, however, about psychosomatic risk factors (e.g., demoralization, hopelessness, vital exhaustion) which have recently emerged as an important source of vulnerability in patients with CVD and HF [16–19] and been linked with increased risk for adverse cardiovascular events and mortality [16, 19–21].

The mechanisms linking such psychosomatic risk factors to HF are largely unknown. Inflammatory activity may be one potential mechanism, given that research shows that heart failure patients have higher levels of proinflammatory cytokines as compared to healthy controls [22–25] and pro-inflammatory cytokines, such as IL-6 and TNF-alpha, have been identified as prognostic markers in HF [26–29]. Particularly, the levels of cytokines (e.g., IL-6, TNF-alpha) have been found higher in in patients with ischemic cardiomyopathy when compared with patients with non-ischemic cardiomyopathy [30].

A growing body of research concerning the association between inflammatory activity and psychosomatic risk factors suggests that: (1) increased levels of cytokines contribute to the development of a constellation of somatic/neurovegetative symptoms collectively referred to as sickness behaviour (e.g., fatigue, lack of energy, anorexia, motor slowing, impaired sleep and decreased motivation) that is clinically close to symptoms of depression and many somatization manifestations [31–35]; (2) elevated levels of proinflammatory cytokines are associated with somatic conditions (e.g., somatization, increased pain sensitivity) [36–38] that are common to multiple clinical conditions including depression [39, 40], eating disorders [41] and somatoform disorders [42–45]; (3) patients treated with cytokines have a higher risk of developing somatization [46] and psychiatric disorders associated with medically unexplained symptoms (e.g., fatigue, irritability, psycho-motor retardation) [47].

The association between pro-inflammatory markers and psychosomatic factors is clinically useful for several reasons such as recognizing subgroups of 'difficult' patients in healthcare settings detecting inexpensive biomarkers potentially predicting the identification of sub-groups of patients with inflammatory activation and at greater risk of adverse outcomes and facilitating the development of individually tailored treatments [48]. However, there is a lack of research studying the association between cytokines and psychosomatic factors in patients with HF [49–51] probably because of difficulty in applying traditional psychiatric diagnostic criteria for somatic symptom disorders to medical patients [52, 53]. The Diagnostic Criteria

for Research in Psychosomatics (DCPR) have been suggested as complementary measures for identifying subthreshold or unclassified psychopathology related to the medical ill [54–56]. They include psychosomatic syndromes related to abnormal illness behaviour (e.g., somatization, hypochondriacal fears and beliefs, and illness denial) and psychological factors (e.g., alexithymia, type A behaviour, demoralization, and irritable mood) [57, 58]. Current emphasis in psychosomatic research on is about analyse differences in outcomes between patients with and without DCPR diagnoses [55, 58, 59]. The objective of this study was to compare heart failure patients with and without DCPR syndromes with respect to the proinflammatory cytokines. Considering the important role that inflammation has in the pathogenesis of psychosomatic disorders, it may follow that the observed associations between psychosomatic syndromes and HF are affected by greater inflammatory activity. Furthermore, given existing evidence that the association between depression and psychological factors with adverse cardiovascular events in HF is partly attributable to inflammation [4, 6, 13, 14], we examined the relation between depression, psychological variables, and the circulating levels of cytokines, in patients with HF. We used serum levels of IL-6 and TNF-alpha as markers of inflammation because multiple lines of evidence indicate that those cytokines have been found to be major prognostic markers in HF [9, 25, 26, 29, 60] and associated with the development of psychosomatic symptoms [40, 51].

## Methods

A cross-sectional study was conducted between May to December 2019 to assess possible correlations between inflammatory cytokines and psychosomatic syndromes in HF patients. Recruitment was interrupted by the onset of the COVID-19 pandemic in early 2020. Consecutive adult HF outpatients referred to the Institute of Cardiology, Azienda Ospedaliera Universitaria Ospedali Riuniti, Foggia, Italy, for routine outpatient visits were recruited. The study was approved by the local Institutional Review Board (9/CE/2016). In accordance with the Declaration of Helsinki, all participants provided written informed consent prior to being included in the study.

### Inclusion/Exclusion criteria

All patients with HF were included. Exclusion criteria were lifetime history of or current psychiatric disorders such as schizophrenia or other psychosis, mental retardation, history of severe head trauma, stroke, neurological disease, alcohol or substance abuse in the past 6 months, being included in the waiting list for cardiac transplantation, and history of a systemic, endocrine, or immune disorders, cancer, infections, and allergies. All subjects were free of immunomodulating drugs including non-steroid anti-inflammatory drugs.

### Testing procedures

All testing procedures were conducted at one day. Baseline information of each patient on demographic factors, personal health habits, medication use, and medical history were collected. Eligible participants completed questionnaires and interview assessment, and blood sample was taken for cytokine measurements. Sociodemographic, anthropometric, smoking status, habitual alcohol intake and clinical data were obtained from medical records. Smoking status had three categories: never, past, and current smoker. Drinking pattern included six categories: lifelong abstainer, ex-drinker, less than once a week, 1–2, 3–5, and 6–7 days/week [61].

## Evaluation of psychological factors

Psychosomatic and psychological factors were evaluated with validated scales. The DCPR is administered during a face-to-face interview and takes about 15–30 min to be completed. It includes items scored in a yes/no response format evaluating the presence of 12 psychosomatic syndromes: a patient may be positive to more than one syndrome at the same time. DCPR clusters of irritable mood, demoralization and persistent somatization were found previously as relevant to HF [18, 19, 55] and were assessed through a semistructured interview [57].

Somatic symptoms were assessed with the PHQ-15, a widely used screening instrument [62]. Patients were requested to rate the severity of their somatic symptoms during the previous 4 weeks on a 3-point scale as either 0 ("not bothered at all"), 1 ("bothered a little") or 2 ("bothered a lot"). Depressive symptoms were measured with the BDI-II yielding, a total score and separate cognitive/affective and somatic/affective subscale scores [63]. Finally, the Psychosocial Index (PSI) was used for assessing for stress and psychosocial dimensions (stress, well-being, psychological distress, abnormal illness behaviour and quality of life) [64]. This self-rating questionnaire includes 55 items. Some questions involve specific responses, most require a yes/no answer, while others are rated on a Likert scale (0–3, from 'not at all' to 'a great deal'). The following domains are covered: a) Stress: this section (items 13–20 and 22–30) is an integration of both perceived and objective stress, life events and chronic stress. It consists of 17 questions with a total score ranging from 0 to 17; b) Well-being: this section (items 31–36) covers different areas of well-being (i.e., positive relations with others (items 31, 32), environmental mastery (items 33, 34) and autonomy (items 35, 36)), with a score ranging from 0 to 6; c) Psychological distress: this section (items 37–51) consists of a checklist of symptoms addressing sleep disturbances, somatization, anxiety, depression and irritability. The total score may range from 0 to 45; d) Abnormal illness behavior: it allows the assessment of hypochondriacal beliefs and bodily preoccupations (items 52–54). The total score may range from 0 to 9; e) Quality of life (item 55): a simple direct question on quality of life is included. The score ranges from 0 to 4.

## Cytokines evaluation

Blood samples were collected by a vacutainer from the subjects' antecubital vein during outpatient clinic hours between 8 a.m. and 5 p.m. following patients' clinic appointment. Blood samples were placed on ice immediately and centrifuged within 30 minutes. Serum samples were stored at -80°C until use. The inflammatory markers were measured using quantitative enzyme-linked immunosorbent assay (ELISA) kits for TNF-alpha (sensitivity: 0.75 pg/ml; RayBiotech, Inc.) and IL-6 (sensitivity: 0.81 pg/ml; Abcam). All tests were measured in accordance with the manufacturer's recommendations. Each sample was run in duplicate, and the average was obtained. Normality of distribution was satisfied for continuous variables except TNF-alpha and IL-6; therefore, these variables were log10 transformed, resulting in normal distributed variables. However, to permit comparison with results from other studies, the cytokine data are presented as the mean±SD of the untransformed data (Table 1).

## Statistical analysis

The characteristics of patients were subjected to an analysis using descriptive analysis. Chi-square were applied for categorical data and t-test for continuous data. DCPR syndromes were recoded as dichotomous variables: DCPR (+)/DCPR (-). DCPR syndromes were recoded as dichotomous variables: DCPR (+)/DCPR (-). Univariate and multivariable regression models (separate models for log-transformed serum levels of each biomarker) were used to evaluate the relationship between inflammatory cytokines (independent variables) and DCPR

**Table 1. Demographic and clinical characteristics of patients (N = 41).**

| Demographic | |
|---|---|
| Age, (mean (yrs) ± SD) | 70.9 ±7.3 |
| Gender (M/F) (N, %) | 34/7 (83/17) |
| Education (mean (yrs) ± SD) | 8.7± 3.6 |
| Having a partner N (%) | 32 (78.0) |
| Clinical | |
| IL-6 pg/ml (mean, ± SD) | 5.36 (4.1) |
| TNF-alpha pg/ml (mean, ± SD) | 1291.4 (2347) |
| Left HF (N, %) | 24 (58.5) |
| Right HF (N, %) | 2 (4.8) |
| Left and Right HF (N, %) | 14 (34.1) |
| HFrEF; EF < 50% N (%) | 34 (82.9) |
| HFpEF; EF > 50% N (%) | 8 (19.5) |
| LVEF % (mean ± SD) | 41 (10.2) |
| NYHA class III N (%) | 16 (39.0) |
| NYHA class II N (%) | 23 (56.0) |
| NYHA class I N (%) | 2 (4.8) |
| Years since HF diagnosis, (mean (yrs) ± SD) | 9.6 ±4.5 |
| Comorbidity N (%) | 29 (70.7) |
| Body mass index, (mean ± SD) | 26.2 (3.3) |
| Smoking habits N (%) | |
| Never | 25 (60.9) |
| Past smokers | 8 (19.5) |
| Current smokers | 8 (19.5) |
| Alcohol intake N (%) | |
| Never | 24 (58.5) |
| Ex-drinker | 1 (2.4) |
| <1 drink week | 0 (0) |
| 1–2 drinks day/week | 15 (36.5) |
| 3–5 drinks day/week | 1 (2.4) |
| 6–7 drinks day/week | 0 (0) |
| DCPR (+) | 23 (56.1%) |
| Medication | |
| ACE-inhibitors N (%) | 15 (36.5) |
| Diuretics N (%) | 38 (92.6) |
| Beta-blockers N (%) | 38 (92.6) |
| Aspirin N (%) | 13 (31.7) |
| Statins N (%) | 39 (95.1) |

syndromes, PHQ-15, Depression, and psychological dimensions (dependent variables). Body-mass index (BMI), New York Heart Association (NYHA) class, left ventricular ejection fraction (LVEF), smoking habits, alcohol consumption, statin use, aspirin use, beta blocker use, age, and gender were used as covariates, in accordance with previous studies concerning their relationship with the inflammatory markers [65]. NYHA class (I–II vs. III; only a small percentage of patients were NYHA class I), gender, smoking status (never-past smokers vs. current smokers) and alcohol consumption (lifelong abstainer vs. 1–2 day/week), statins and aspirin use were recoded into dichotomous variables. The level of significant difference was set at $p < .05$.

## Results

In total 63 outpatients were invited to participate, and 21 (33.3%) declined. The most cited reason was a lack of time. One subject was excluded from the final sample because the levels of cytokines were rated as outliers (> 2 standard deviations compared to the overall group average). The sample, therefore, comprised 41 patients. Results for demographic and clinical variables are in the Table 1. Continuous variables are expressed as mean ± SD. All patients had ischemic aetiology of HF. Patients were grouped based on their left ventricular ejection fraction (LVEF) record into two: HF patients with reduced ejection fraction HFpEF (LVEF ≥ 50%) and HF patients with preserved ejection fraction HFrEF (LVEF < 50%) [66] (Table 1).

The small fraction (N = 4) of undetectable cytokine concentration values were replaced with the minimum detectable (i.e., sensitivity) value, as is standard practice [67]. There were no significant differences in terms of demographic characteristics between the participants and the patients who refused to participate. Most patients (N = 29, 70.7%) had at least one co-occurring medical condition, mostly arterial hypertension (80.4%), dyslipidaemia (78.0%), diabetes (48.7%), chronic renal failure (31.7) and chronic obstructive pulmonary disease (26.8%). At least one DCPR syndrome was found in 23 (56.1%) patients. Persistent somatization was found in 13 (31.7%) patients, demoralization in 10 (24.3%), and irritable mood in 3 (7.3%). Multiple psychosomatic diagnoses (DCPR>1) were found in 6 (14.6%). Patient characteristics, by DCPR status, are presented in Table 2.

There was no difference in patients who received at least a DCPR diagnosis as compared to patients without DCPR diagnoses in terms of age, gender, LVEF, HF phenotypes (left HF, right HF, both sides HF; HFrEF, HFpEF), smoking status, alcohol intake, statins use aspirin use, beta blockers use. Significantly higher scores were found on depression (BDI-II total score), PSI psychological distress and PHQ-15 somatization in patients presenting with at least one DCPR diagnosis compared to those who did not.

Table 3 summarizes regression analyses of the relationship between inflammatory markers and psychological and psychosomatic risk factors. After controlling for possible confounding factors (BDI-II scores, LVEF, BMI, NYHA class, smoking habits, alcohol consumption, statin use, aspirin use, beta blockers use, age and gender), significant relationships were found between IL-6 levels and psychosomatic diagnoses and symptoms of somatization. Other correlations remained nonsignificant.

## Discussion

This exploratory study presents preliminary results suggesting a possible psychoneuroimmune link between psychosomatic psychopathology and HF. We showed for the first time, to our knowledge, that HF patients who received at least one of the three DCPR diagnoses had significantly higher levels of the proinflammatory cytokine IL-6, but not TNF-alpha, compared with patients with no DCPR diagnoses. Multivariate regression analyses showed that after adjustment for sociodemographic and clinical variables proinflammatory cytokine IL-6 shows the strongest associations with DCPR diagnoses and PHQ-15 somatization.

Our results are in agreement with previous studies suggesting that pro-inflammatory cytokines are involved in the pathophysiology of psychosomatic psychopathology (i.e., vital exhaustion) in patients with coronary heart disease (CHD) [49, 68]. The findings in patients with CHD are relevant to our patients since they had ischemic aetiology of HF and the two conditions (HF and CHD) often share similar characteristics [69]. Growing evidence suggests an association between depressive symptomatology and pro-inflammatory cytokines among patients with HF [6, 13, 65]. However, our study demonstrated an association between IL-6

**Table 2. Between-group comparison of psychological measures, and inflammatory markers.** Mean (SD).

| | DCPR (+) N = 23 | DCPR (-) N = 18 | t-test/ χ2 | P-values |
|---|---|---|---|---|
| Age (years) | 71.0 (7.1) | 70.7 (7.8) | t = 0.15 | 0.87 |
| Gender (F/M) | 5/18 | 2/16 | χ2 = 0.37 | 0.32 |
| Left HF (N, %) | 13/23 (56.5) | 11/18 (61.1) | χ2 = 0.87 | 0.54 |
| Right HF (N, %) | 2/23 (8.6) | 0/18 (0.0) | χ2 = 0.21 | 0.33 |
| Left and Right HF (N, %) | 8/23 (34.7) | 7/18 (38.8) | χ2 = 0.85 | 0.54 |
| HFrEF; EF < 50% N (%) | 17/23 (73.9) | 16/18 (88.8) | χ2 = 0.69 | 0.43 |
| HFpEF; EF > 50% N (%) | 5/23 (21.7) | 2/18 (11.1) | χ2 = 0.44 | 0.37 |
| LVEF (mean ± SD) | 41.4 (11.1) | 41.3 (8.9) | t = 0.03 | 0.97 |
| BMI (kg/m$^2$) | 26.7 (2.3) | 25.0 (4.4) | t = 1.61 | 0.11 |
| BDI-II total | 16.0 (6.0) | 11.3 (8.3) | t = 2.04 | 0.04 |
| BDI-II somatic | 8.3 (4.1) | 6.2 (4.0) | t = 1.56 | 0.12 |
| BDI-II cognitive/affective | 7.6 (3.7) | 5.1 (4.8) | t = 1.92 | 0.06 |
| PHQ-15 | 12.0 (2.9) | 5.5 (2.3) | t = 7.68 | < 0.001 |
| PSI stress | 1.7 (2.1) | 1.4 (1.7) | t = 0.54 | 0.59 |
| PSI psychological distress | 14.5 (10) | 5.3 (4.7) | t = 3.52 | 0.001 |
| PSI abnormal illness behavior | 0.3 (0.7) | 0.2 (0.7) | t = 0.52 | 0.60 |
| PSI well-being | 3.8 (2.2) | 3.5 (2.6) | t = 0.35 | 0.72 |
| PSI quality of life | 1.4 (1.0) | 1.6 (1.4) | t = -0.48 | 0.63 |
| Smoking status | 18/5 | 15/3 | χ2 = 0.02 | 0.54 |
| Alcohol intake | 12/11 | 12/6 | χ2 = 0.23 | 0.41 |
| Statins use | 14/9 | 15/3 | χ2 = 0.42 | 0.34 |
| Aspirin use | 7/16 | 6/12 | χ2 = 0.02 | 0.56 |
| Beta blockers | 21/2 | 17/1 | χ2 = 0.01 | 0.56 |
| IL-6 (log, mean ± SD) | 0.72 (0.30) | 0.43 (0.34) | t = 2.87 | 0.006 |
| TNF-alpha (log, mean ± SD) | 1.86 (0.94) | 2.46 (1.1) | t = -1.84 | 0.07 |

DCPR = Diagnostic Criteria for Research in Psychosomatics; psychosomatic syndromes were reported as dichotomous variables: (e.g., DCPR (+) / DCPR (-));

HFrEF = Heart Failure with Reduced Ejection Fraction; HFpEF = Heart Failure with Preserved Ejection Fraction; LVEF = left ventricular ejection fraction;

BDI-II = Beck Depression Inventory-II; PHQ-15 = Patient Health Questionnaire-15; PSI = Psychosocial Index: all these scores were reported as continuous variables;

smoking status was reported as dychotomus variables (never-past smokers/current smokers); alcohol intake reported as dychotomus variables (users/no-users).

levels and psychosomatic syndromes while there was no such association with depression. These findings reflect the study by Jansky et al. [49] which found a significant relation between IL-6 levels and psychosomatic psychopathology (i.e., vital exhaustion), but no depression, in patients with coronary heart disease. Our results are also in agreement with several studies suggesting that IL-6 is involved in the pathophysiology of psychosomatic disorders [45] and fibromyalgia [70] independently of comorbid medical conditions including depression.

Several lines of research suggest that proinflammatory cytokines including IL-6 may play an important role in regulating normal physiological processes such as learning and memory [71]. In contrast, chronically elevated IL-6 above physiological levels may alter neuronal plasticity and may be important for the pathogenesis of psychopathology [14, 15]. Our findings lend support to the hypothesis of the association between psychosomatic diagnoses and the pro-inflammatory cytokine among patients with HF. Our findings also suggest the possibility that elevated IL-6 levels are more relevant for the pathogenesis of psychosomatic syndromes than for depression in patients with HF. There should be a particular focus on DCPR persistent somatization which was found in 30% of our sample. Furthermore, significant relationships were found between IL-6 levels and symptoms of somatization (PHQ-15). This is in line

**Table 3. Linear relation between inflammatory markers and psychosomatic syndromes, and psychological dimensions.**

| | IL-6 | | | TNF-alpha | | |
|---|---|---|---|---|---|---|
| | β | 95%CL | p value | β | 95%CL | p value |
| Model 1: psychosomatic syndromes and psychological dimensions | | | | | | |
| DCPR syndromes (+) | 0.41 | 0.12,0.71 | 0.006 | -0.28 | -0.59,0.02 | 0.07 |
| PHQ-15 | 0.42 | 0.13,0.71 | 0.005 | -0.30 | -0.61,0.006 | 0.06 |
| Depression | | | | | | |
| BDI total score | 0.23 | -0.08,0.54 | 0.13 | 0.10 | -0.21,0.43 | 0.49 |
| BDI cognitive score | 0.18 | -0.13,0.50 | 0.24 | 0.09 | -0.22,0.41 | 0.55 |
| BDI somatic score | 0.21 | -0.09,0.53 | 0.16 | 0.09 | -0.22,0.41 | 0.56 |
| Stress | -0.18 | -0.49,0.13 | 0.25 | -0.06 | -0.38,0.26 | 0.70 |
| Well-being | -0.23 | -0.54,0.08 | 0.14 | -0.03 | -0.36,0.28 | 0.81 |
| Psychological distress | 0.03 | -0.29,0.35 | 0.84 | -0.13 | -0.45,0.18 | 0.40 |
| Abnormal illness behaviour | 0.19 | -0.12,0.51 | 0.21 | -0.15 | -0.47,0.15 | 0.31 |
| Quality of life | -0.13 | -0.45,0.18 | 0.39 | -0.13 | -0.45,0.18 | 0.38 |
| Model 2: psychosomatic syndromes adjusted for covariates* | | | | | | |
| DCPR syndromes (+) | 0.44 | 0.01,0.88 | 0.04 | -0.11 | -0.60,0.37 | 0.63 |
| PHQ-15 | 0.60 | 0.21,0.92 | 0.003 | -0.24 | -0.68,0.19 | 0.26 |

*Adjusted for Body-mass index (BMI), LVEF = left ventricular ejection fraction, New York Heart Association

(NYHA) class, smoking habits, statin use, aspirin use, alcohol consumption, beta blockers use, age, and gender.

DCPR syndromes were reported as dichotomous variables. β = standardized regression coefficient.

CL = Confidence Limits.

with recent findings on the role of IL-6 cytokine in the pathogenesis of psychosomatic psychopathology.

Proinflammatory cytokines, notably IL-6, are responsible for the selective activation of indoleamine 2,3 dioxygenase (IDO) which, in turn, leads both to lower levels of tryptophan, the essential amino acid precursor of serotonin, and an increased neurotoxic tryptophan catabolites (TRYCATs) [72]. Interestingly, Maes et al. [73] demonstrated that somatization, but not depression, is characterized by increased IDO activity and disorders in the tryptophan catabolite pathway. Therefore, it is conceivable that IL-6 plays a role in the development of symptoms of somatization through effects on IDO activity and serotoninergic neurotransmission. In line with this hypothesis previous studies have indicated that proinflammatory cytokines have an important role in the pathophysiology of somatosensory amplification [38, 74].

Existing evidence also suggests that immunological processes may potentially mediate and moderate the association between psychological factors (e.g., subjective well-being, chronic psychological distress) and cardiovascular disease risk [6, 49]. It is not clear why in our study similar relations between psychological dimensions and levels of inflammatory markers were not found. It is probable the limited sample size might have played a role in the lack of significant findings. A larger sample of patients might have revealed significant associations. However, it is still not known whether subjective well-being and psychological distress affect inflammation, or vice versa. Further studies are needed to address the issue of a possible psychoneuroimmune link between psychological factors and cardiovascular disease. A significant association was found between the DCPR-diagnosis and PSI psychological distress. This is in accordance with a previous study that found a strong association between the DCPR syndromes and increased levels of psychological distress in patients with chronic cardiovascular diseases [55]. These findings are in accordance with increased allostatic load (physiological

burden) in individuals with psychiatric/psychological diagnoses and represent further support for the validity of DCPR diagnoses in detecting exposure to environmental challenges that exceed the resilience resources of subjects.

A number of limitations in the study deserve consideration. First, the limited sample size, the imbalance of male and female participants and the single-center design means that firm conclusions cannot be drawn, and further investigation is required in a large study or a multi-center trial. Second, its cross-sectional design fails to allow a longitudinal assessment of the stability of psychosomatic diagnoses. Third, it is also possible that residual confounding, unrelated to factors included in our regression analyses, could be related to IL-6 levels. Fourth, the lack of association between depression and inflammatory markers could be attributed to our method measuring depression, and we cannot rule out the possibility that using other questionnaires than the BDI-II (e.g., HAMD) would lead to different results. Fifth, it is well known that many mechanisms may mediate the effect of chronic mental stress on inflammation and atherosclerosis. They include the hypothalamic-pituitary-adrenal and individual factors related to personality. Therefore, we cannot exclude that these factors and their interaction with inflammatory processes may have contributed to the development of psychosomatic symptom dimensions related to inflammation. Further research is required to explore the possible role of cytokines on HF and the extent to which such an association is mediated or moderated by specific modulatory factors. Finally, increasing evidence suggests that functioning of the immune system exhibit peculiar circadian rhythms that are synchronized and coordinated by the suprachiasmatic nucleus (SCN). Consequently, the diurnal variation of inflammatory cytokines poses a risk of confounding if sampling is performed without regard to time of day. Particularly, the production of the pro-inflammatory cytokines, including interleukin IL-6 and TNF-alpha, exhibit diurnal rhythmicity that correlates inversely with plasma cortisol [75, 76]. Indeed, earlier studies on diurnal variation of cytokines reported that cytokine levels were highest in blood taken in the early morning or late evening [77, 78]. In this study blood samples were collected between 8 a.m. and 5 p.m. as in the study by Kupper et [65]. However, we cannot rule out the possibility that our findings have been influenced by the diurnal variation of inflammatory cytokines. Despite these limitations, our findings indicate the necessity to broaden the assessment targets of psychiatric evaluation in HF patients and substantiate the validity and clinical utility of the DCPR in detecting clinical prognostic factors that are not detected using traditional psychiatric taxonomy. In conclusion, results suggest a possible psychoneuroimmune link between psychosomatic psychopathology and HF. This information may provide clinicians with more accurate data that can be used to successfully manage such patients. Further studies are recommended to replicate these findings and to study the relationship between psychosomatic dimensions and inflammation in HF.

## Supporting information

**S1 Dataset.**
(XLSX)

## Acknowledgments

The authors would like to thank dr. Correale Michele, dr. Alfieri Simona and dr. Elia Antonella for their help with participants recruitment.

## Author Contributions

**Conceptualization:** Mario Altamura, Piero Porcelli.

**Data curation:** Eleonora Angelini, Fabio M. P. Tortorelli, Angela Balzotti.

**Formal analysis:** Giovanna D'Andrea.

**Investigation:** Eleonora Angelini, Fabio M. P. Tortorelli, Angela Balzotti, Maurizio Margaglione, Natale D. Brunetti, Tommaso Cassano.

**Methodology:** Giovanna D'Andrea.

**Resources:** Maurizio Margaglione.

**Supervision:** Mario Altamura, Piero Porcelli, Natale D. Brunetti, Tommaso Cassano, Antonello Bellomo.

**Writing – original draft:** Mario Altamura.

**Writing – review & editing:** Mario Altamura.

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
