## [Decision Letter · Decision Letter 0]

3 Jan 2022

PONE-D-21-35033Psychosomatic syndromes are associated with IL-6 pro-inflammatory cytokine in heart failure patients.PLOS ONE

Dear Dr. Altamura,

Thank you for submitting your manuscript to PLOS ONE. After careful consideration, we feel that it has merit but does not fully meet PLOS ONE’s publication criteria as it currently stands. Therefore, we invite you to submit a revised version of the manuscript that addresses the points raised during the review process.

The two reviewers addressed several major and minor concerns about your manuscript. Please revise your manuscript carefully.

We look forward to receiving your revised manuscript.

Kind regards,

Kenji Hashimoto, PhD

Academic Editor

PLOS ONE

Reviewers' comments:

Reviewer's Responses to Questions

**Comments to the Author**

1. Is the manuscript technically sound, and do the data support the conclusions?

Reviewer #1: Yes

Reviewer #2: Partly

2. Has the statistical analysis been performed appropriately and rigorously? 

Reviewer #1: Yes

Reviewer #2: No

3. Have the authors made all data underlying the findings in their manuscript fully available?

Reviewer #1: Yes

Reviewer #2: Yes

4. Is the manuscript presented in an intelligible fashion and written in standard English?

Reviewer #1: Yes

Reviewer #2: Yes

5. Review Comments to the Author

Reviewer #1: A cross-sectional study on patients with heart failure (HF) was conducted to investigate the association between proinflammatory cytokines and psychosomatic psychopathology in the Institute of Cardiology at Azienda Ospedaliera Universitaria Ospedali Riuniti. The main clinical findings of this study showed that IL-6 level is correlated with psychosomatic psychopathology but no depression by DCPR status and other measure indactors. The advantage of this article is a consecutive study and innovation idea in clinical study, although the COVID-19 pandemic. However, small sample size will weaken the proof of this paper. Meanwhile, two issues should be addressed.

1. As we know, many pro-inflammatory cytokines, such as IFNγ, IL-1β and IL-6, might communicate with the central nervous system to stimulate an immune response in the brain, which may cause or exacerbate psychological symptoms. Whether only IL-6 and TNF-alpha are the key cytokines in this study of patients? Measuring panels of cytokines may contribute to research in support of a psychoneuroimmune connection between psychosomatic psychopathology and HF in patients. However, in this manuscript, the author just measured IL-6 and TNF-alpha, please discuss the reason.

2. There is still a question that the authors stated the view that IL-6 is not related to the depression, assessed by depression scores from BDI-II. Measure indicator adopted by the researcher will influence the conclusion. BDI-II is better for this study, or other measure indicator like HAMD is better? After all, strict criteria contribute to the validity of the study.

Reviewer #2: The authors provide an explorative cross-sectional study (n=41) that focuses on cytokines' influence on psychosomatic syndromes development in heart failure patients. Authors evaluated psychosomatic syndromes by different questioners and searched for an association between psychosomatic syndromes and IL-6 and TNF-alpha.

My main remarks are:

1. The study focuses on HF patients. However, a clear description of this specific cohort is missing, while HF is a multi-etiologic syndrome with different phenotypes and disease stages. Based on the information given, it is hard to understand what kind of HF patients were enrolled in the study. I would recommend extending baseline characteristics by describing the HF group in more detail. It is important to know the etiology of HF (at least the main causes) and phenotypes (was it left or right HF? HFpEF or HFrEF?) for several reasons.

a. It is known that HF is a proinflammatory state. However, levels of cytokines differ between different etiologies (e.g., ischemic vs. non-ischemic) (https://doi.org/10.2478/s11536-013-0233-y. As well, cytokines play a role in the pathogenesis and development of HF. Cytokines concentration is positively associated with parameters of HF severity (DOI: 10.1007/s11845-017-1680-2, DOI: 10.3390/life11101006, DOI: 10.1007/s11845-017-1680-2).

b. To my knowledge, the profile of psychosocial disorders differs between distinct groups of cardiovascular diseases. In addition, the incidence of depression increases with HF severity. Knowing more about the cohort would also aid in interpreting the results of the prevalence of psychosomatic syndromes and even their relationship with cytokines.

c. In the discussion section, the authors compare their findings with previous studies. Still, it remains unclear whether the study patients (with HF) are similar to previous ones or differ significantly.

Therefore, I think a more detailed definition of the HF cohort would help get a full view and lead to more insights into the link between HF, cytokines, and psychosomatic syndromes.

2. I would recommend that the authors clearly state the aim of the study.

3. Statistics description states that cytokines were dependent variables and various psychosomatic scores – independent ones. That would lead to the hypothesis that the authors are searching how cytokines' levels (outcome variable) depend on psychosomatic syndromes. However, I presume that the aim was the opposite. Therefore, I would recommend revising the statistics description and statistical analysis (or clarifying the aim of the study).

4. In the statistics description section, it is written that IL-6 and TNF-alpha were checked for normality. I wonder whether the authors checked normality for other continuous variables, presented in table 2. Also, I wonder if there are just two groups (DCPR+ and DCPR-), why did the authors choose ANOVA for analysis instead of a t-test? An explanation of why patients were stratified by DCPR results would be helpful.

5. How are continuous variables expressed (mean ± SD, median (IQR), or else)? I could not find that neither in the statistics description.

6. There is no mention what were the levels of cytokines. The values of cytokines should be added in Table 1 (measured values, not logarithmic ones, so it would be easier to understand if it is normal or elevated). There is written in the discussion section that "increased inflammatory activity" (line 274), but the proof is missing in the results section.

7. It is unclear what the authors mean by "comorbidity" in table 1? I presume that the conditions mentioned in lines 187-190, even the case numbers do not match (n=29 in the table and n=27 in the text). It needs clarification in the table. In addition, what does it mean systemic disorders (line 102)?

8. Some of the results are presented as text (lines 219-225). It would be easier to follow them if they were at the table.

9. I think that it would be easier to read if the article would be divided into subsections (e.g., methods: inclusion/exclusion criteria; cytokines evaluation; evaluation of psychosomatic and psychological factors, etc.), at least in separate paragraphs. In addition, I would recommend extending the psychosomatic evaluation part and describing in more detail what was evaluated, how it was done, and the ranges of scores of each questionnaire.

10. I think that other statistical methods such as ROC analysis could also fit this study (just an idea for authors).

6. PLOS authors have the option to publish the peer review history of their article (what does this mean?). If published, this will include your full peer review and any attached files.

Reviewer #1: **Yes: **Yan Wei, Southwest medical university

Reviewer #2: No

---

## [Author Response · Author response to Decision Letter 0]

17 Feb 2022

Reviewer #1: A cross-sectional study on patients with heart failure (HF) was conducted to investigate the association between proinflammatory cytokines and psychosomatic psychopathology in the Institute of Cardiology at Azienda Ospedaliera Universitaria Ospedali Riuniti. The main clinical findings of this study showed that IL-6 level is correlated with psychosomatic psychopathology but no depression by DCPR status and other measure indactors. The advantage of this article is a consecutive study and innovation idea in clinical study, although the COVID-19 pandemic. However, small sample size will weaken the proof of this paper. Meanwhile, two issues should be addressed.

1. As we know, many pro-inflammatory cytokines, such as IFNγ, IL-1β and IL-6, might communicate with the central nervous system to stimulate an immune response in the brain, which may cause or exacerbate psychological symptoms. Whether only IL-6 and TNF-alpha are the key cytokines in this study of patients? Measuring panels of cytokines may contribute to research in support of a psychoneuroimmune connection between psychosomatic psychopathology and HF in patients. However, in this manuscript, the author just measured IL-6 and TNF-alpha, please discuss the reason.

Thank you very much prof. Yan Wei for the positive comments and careful review, which helped improve the manuscript. A variety of inflammatory markers have been identified as having a potential role in the progression of HF. However, multiple lines of evidence indicate that IL-6 and TN-alpha cytokines have been found to be major prognostic markers in HF (https://pubmed.ncbi.nlm.nih.gov/11319194/;
https://pubmed.ncbi.nlm.nih.gov/17170586/;
https://pubmed.ncbi.nlm.nih.gov/10808148/;
https://pubmed.ncbi.nlm.nih.gov/31087601/ ; https://pubmed.ncbi.nlm.nih.gov/28889349/;
https://pubmed.ncbi.nlm.nih.gov/21263314/ ). Other cytokines may also play a role in the development of HF, but their precise roles are not well understood. 

An extensive literature suggests that psychological risk factors (e.g., depression) play a critical role in the progression HF and a growing body of evidence indicate that the association of those psychological risk factors with adverse cardiovascular events is due to immunological processes, specifically inflammation (Kop WJ, Mommersteeg PMC. Psychoneuroimmunological processes in coronary artery disease and heart failure. In: Alexander W. Kusnecov AW, Anisman H, editors. The Wiley-Blackwell Handbook of Psychoneuroimmunology. 2013. pp. 504-523; Silk J, Volker A. Does inflammation link clinical depression and coronary artery disease? In: Baune T.B. editor. Inflammation and immunity in depression. https://www.ncbi.nlm.nih.gov/pmc/articles/PMC3059072/). Many studies have examined the association between serum levels of a broad range of cytokines and psychological risk factors in cardiovascular disease. However, recent investigations that address association between inflammatory biomarkers and psychological risk factors (e.g., depression, psychosomatic symptoms) in HF exhibit the most consistent associations for IL-6 and TNF-alpha (https://pubmed.ncbi.nlm.nih.gov/16084159/ ; https://pubmed.ncbi.nlm.nih.gov/23399050/ ; https://pubmed.ncbi.nlm.nih.gov/31615595/ ). We have now added in the introduction section the following sentence: “We used serum levels of IL-6 and TNF-alpha as markers of inflammation because multiple lines of evidence indicate that those cytokines have been found to be major prognostic markers in HF [9, 25, 26, 29, 61] and associated with psychosomatic symptoms [40, 51].”

2. There is still a question that the authors stated the view that IL-6 is not related to the depression, assessed by depression scores from BDI-II. Measure indicator adopted by the researcher will influence the conclusion. BDI-II is better for this study, or other measure indicator like HAMD is better? After all, strict criteria contribute to the validity of the study.

We thank the reviewer for this constructive comment. We have added in the Discussion section the following sentence: “The lack of association between depression and inflammatory markers could be attributed to our method measuring depression, and we cannot rule out the possibility that using other questionnaires than the BDI-II (e.g., HAMD) would lead to different results”.

Reviewer #2: The authors provide an explorative cross-sectional study (n=41) that focuses on cytokines' influence on psychosomatic syndromes development in heart failure patients. Authors evaluated psychosomatic syndromes by different questioners and searched for an association between psychosomatic syndromes and IL-6 and TNF-alpha.

My main remarks are:

The study focuses on HF patients. However, a clear description of this specific cohort is missing, while HF is a multi-etiologic syndrome with different phenotypes and disease stages. Based on the information given, it is hard to understand what kind of HF patients were enrolled in the study. I would recommend extending baseline characteristics by describing the HF group in more detail. It is important to know the etiology of HF (at least the main causes) and phenotypes (was it left or right HF? HFpEF or HFrEF?) for several reasons.

We have now revised the manuscript according to the reviewer’s suggestions and reported this information in the Table 1 and in the full text. All patients had ischemic aetiology of HF. 24 patients (58.5%) had left heart failure; 14 patients (34.1%) had heart failure on both sides and 2 patients (4.8%) had right heart failure. Patients were grouped based on their LVEF record into two: patients with HFpEF (LVEF ≥ 50%) and patients with HFrEF (LVEF < 50%). https://www.ncbi.nlm.nih.gov/pmc/articles/PMC5117494/. 34 patients (82.9%) had HF with reduced ejection fraction (HFrEF; EF < 50%); and 8 patients (19.5%) had preserved ejection fraction (HFpEF; EF ≥ 50%). Mean left ventricular ejection fraction (LVEF) was 41.0 ± 10.2%. There was no difference in patients who received at least a DCPR diagnosis as compared to patients without DCPR diagnoses in terms of LVEF, HF phenotypes (left HF, right HF, both sides HF; HFrEF, HFpEF). Furthermore, when LVEF was entered in the multivariable regression model, the association between IL-6 levels and psychosomatic diagnoses and symptoms of somatization remained virtually unchanged.

Tab.1 Demographic and clinical characteristics of patients (N=41).

Demographic

Age, (mean (yrs) ± SD) 70.9 ±7.3

Gender (M/F) (N, %) 34/7 (83/17)

Education (mean (yrs) ± SD) 8.7± 3.6

Having a partner N (%) 32 (78.0)

Clinical

IL-6 pg/ml (mean, ± SD) 5.36 (4.1)

TNF-alpha pg/ml (mean, ± SD) 1291.4 (2347)

Left HF (N, %) 24 (58.5)

Right HF (N, %) 2 (4.8)

Left and Right HF (N, %) 14 (34.1) 

HFrEF; EF < 50% N (%) 34 (82.9)

HFpEF; EF > 50% N (%) 8 (19.5)

LVEF % (mean ± SD ) 41 (10.2)

NYHA class III N (%) 16 (39.0)

NYHA class II N (%) 23 (56.0)

NYHA class I N (%) 2 (4.8)

Years since HF diagnosis,

(mean (yrs) ± SD) 9.6 ±4.5

Comorbidity N (%) 29 (70.7)

Body mass index, (mean ± SD) 26.2 (3.3)

Smoking habits N (%)

 Never 25 (60.9)

 Past smokers 8 (19.5)

 Current smokers 8 (19.5)

Alcohol intake N (%)

 Never 24 (58.5)

 Ex-drinker 1 (2.4)

 <1 drink week 0 (0)

 1–2 drinks day/week 15 (36.5)

 3–5 drinks day/week 1 (2.4)

 6-7 drinks day/week 0 (0)

DCPR (+) 23 (56.1%) 

Medication

ACE-inhibitors N (%) 15 (36.5)

Diuretics N (%) 38 (92.6)

Beta-blockers N (%) 38 (92.6)

Aspirin N (%) 13 (31.7)

Statins N (%) 39 (95.1)

a. It is known that HF is a proinflammatory state. However, levels of cytokines differ between different etiologies (e.g., ischemic vs. non-ischemic) (https://doi.org/10.2478/s11536-013-0233-y. As well, cytokines play a role in the pathogenesis and development of HF. Cytokines concentration is positively associated with parameters of HF severity (DOI: 10.1007/s11845-017-1680-2, DOI: 10.3390/life11101006, DOI: 10.1007/s11845-017-1680-2).

We agree with the reviewer that levels of cytokines differ between ischemic vs non-ischemic HF and thank her/him for the references which will be included in the revised version of the manuscript. We have now characterized our sample with regards to aetiology of HF and Left ventricular ejection fraction (LVEF) and revised the manuscript according to the reviewer’s comments. In the Introduction section: 

” A substantial number of clinical studies have clearly established the involvement of cytokines in the pathophysiology of major depression and ischemic CVD with and without comorbid depression [6, 7, 8]. Evidence shows that proinflammatory cytokines are important determinants of severity and prognosis of CVD [9, 10, 11] (https://pubmed.ncbi.nlm.nih.gov/28889349/ ; https://pubmed.ncbi.nlm.nih.gov/30259192/ ). Plasma levels of cytokines are strong predictors of patients’ functional class (NYHA) and also predictors of left ventricular function (LVEF) decline [12]. Furthermore, and maybe even more important, a growing body of evidence indicate that the association of psychological risk factors with adverse cardiovascular events is due to immunological processes, specifically inflammation [4, 6, 7, 13, 14, 15].

b. To my knowledge, the profile of psychosocial disorders differs between distinct groups of cardiovascular diseases. In addition, the incidence of depression increases with HF severity. Knowing more about the cohort would also aid in interpreting the results of the prevalence of psychosomatic syndromes and even their relationship with cytokines.

We have now revised the introduction according to the reviewer's comments and added the following sentences in the Introduction section. 

“An extensive literature suggests that psychological risk factors (e.g., depression, negative personality traits) play a critical role in the progression and mortality of cardiovascular disease (CVD) and heart failure (HF) [1, 2]. Evidence suggests that depressive symptoms might be associated with adverse outcomes and disease severity in patients with HF, independently of biomedical risk factors [2, 3, 4]” (https://pubmed.ncbi.nlm.nih.gov/18474348/ ; https://pubmed.ncbi.nlm.nih.gov/29975336/). 

We certainly agree with the reviewer that psychosocial risk factors differ between different diagnostic groups of cardiovascular diseases. 

“The profile of psychosocial risk factors for CVD revealed relevant variability between different diagnostic groups of cardiovascular diseases. For instance, Stauber et al., 2012 ( https://pubmed.ncbi.nlm.nih.gov/22426505/ ) compared psychosocial risk factors for CVD across the affective spectrum (ie, depression, anxiety, vital exhaustion, positive affect), personality characteristics (ie, hostility, type D personality), and social support between 3 groups of patients with a primary diagnosis of CHD, CHF, or PAD. The authors reported that relative to patients with peripheral arterial disease, those with HF showed greater exhaustion and lower positive affect. This clearly shows the relative importance of psychosomatic risk factors (e.g., vital exhaustion) versus other variables in patients with HF and it is in line with recent evidence that suggests that psychosomatic risk factors are an important source of vulnerability in patients with HF .” 

c. In the discussion section, the authors compare their findings with previous studies. Still, it remains unclear whether the study patients (with HF) are similar to previous ones or differ significantly. Therefore, I think a more detailed definition of the HF cohort would help get a full view and lead to more insights into the link between HF, cytokines, and psychosomatic syndromes.

We have now provided more details about the study sample (see Tab. 1)

We have now added the following sentence in the Discussion section: “Our results are in agreement with previous studies suggesting that pro-inflammatory cytokines are involved in the pathophysiology of psychosomatic psychopathology (i.e., vital exhaustion) in patients with coronary heart disease (CHD) [49, 69] ( https://pubmed.ncbi.nlm.nih.gov/23524631/ ). The findings in patients with CHD are relevant to our patients since they had ischemic aetiology of HF and the two conditions (HF and CHD) often share similar characteristics [70]. “

2. I would recommend that the authors clearly state the aim of the study.

We have now clarified the aim of the study in the Introduction. “The objective of this study was to compare heart failure patients with and without DCPR syndromes with respect to the proinflammatory cytokines. Considering the important role that inflammation has in the pathogenesis of psychosomatic disorders, it may follow that the observed associations between psychosomatic syndromes and HF are affected by greater inflammatory activity. Furthermore, given existing evidence that the association between depression and psychological factors with adverse cardiovascular events in HF is partly attributable to inflammation [4, 6, 13, 14], we examined the relation between depression, psychological variables, and the circulating levels of cytokines, in patients with HF.”

3. Statistics description states that cytokines were dependent variables and various psychosomatic scores – independent ones. That would lead to the hypothesis that the authors are searching how cytokines' levels (outcome variable) depend on psychosomatic syndromes. However, I presume that the aim was the opposite. Therefore, I would recommend revising the statistics description and statistical analysis (or clarifying the aim of the study).

We have now revised the manuscript according to the reviewer’s suggestions. We have now used univariate and multivariate regression models with cytokines as independent variables and psychosomatic syndromes, psychological dimensions, and Depression as dependent variables:

“Univariate and multivariable regression models (separate models for log-transformed serum levels of each biomarker) were used to evaluate the relationship between inflammatory cytokines (independent variables) and DCPR syndromes, PHQ-15, Depression, and psychological dimensions (dependent variables)”

Tab. 3 

 IL-6 TNF-alpha 

 β 95%CL p value β 95%CL p value 

Model 1 : psychosomatic syndromes and psychological dimensions

DCPR syndromes (+) 0.41 0.12,0.71 0.006 -0.28 -0.59,0.02 0.07

PHQ-15 0.42 0.13,0.71 0.005 -0.30 -0.61,0.006 0.06

Depression 

BDI total score 0.25 -0.06,0.56 0.11 0.13 -0.18,0.45 0.40

BDI cognitive score 0.19 -0.15,0.51 0.22 -0.005 -0.32,0.31 0.97 

BDI somatic score 0.29 -0.01,0.60 0.06 0.08 -0.23,0.41 0.58 

Stress -0.18 -0.49,0.13 0.25 -0.06 -0.38,0.26 0.70

Well-being -0.23 -0.54,0.08 0.14 -0.03 -0.36,0.28 0.81

Psychological distress 0.03 -0.29,0.35 0.84 -0.13 -0.45,0.18 0.40 

Abnormal illness behaviour 0.19 -0.12,0.51 0.21 -0.15 -0.47,0.15 0.31

Quality of life -0.13 -0.45,0.18 0.39 -0.13 -0.45,0.18 0.38

Model 2 : psychosomatic syndromes adjusted for covariates*

DCPR syndromes (+) 0.44 0.01,0.88 0.04 -0.11 -0.60,0.37 0.63

PHQ-15 0.60 0.21,0.92 0.003 -0.24 -0.68,0.19 0.26

*Adjusted for Body-mass index (BMI), LVEF= left ventricular ejection fraction, New York Heart Association 

(NYHA) class, smoking habits, statin use, aspirin use, alcohol consumption, beta blockers use, age, and gender. 

DCPR syndromes were reported as dichotomous variables. β = standardized regression coefficient. 

CL = Confidence Limits.

4. In the statistics description section, it is written that IL-6 and TNF-alpha were checked for normality. I wonder whether the authors checked normality for other continuous variables, presented in table 2. Also, I wonder if there are just two groups (DCPR+ and DCPR-), why did the authors choose ANOVA for analysis instead of a t-test? An explanation of why patients were stratified by DCPR results would be helpful.

We have clarified this issues in the cytokines evaluation section: “Normality of distribution was satisfied for continuous variables except TNF-alpha and IL-6; therefore, these variables were log10 transformed. However, to permit comparison with results from other studies, the cytokine data are presented as the mean±SD of the untransformed data (Tab.1).”

The statistical analyses have been redone according to reviewer instructions, using t-test. 

The DCPR were developed by an international group of investigators to translate into operational tools 12 psychosomatic syndromes that were found to have a prognostic value in the development and outcome of various medical diseases. We have added the following sentence in Introduction section: “Current emphasis in psychiatry is about analyse differences in outcomes between patients with and without DCPR diagnoses with respect to the proinflammatory cytokines. ” (Bellomo et al. 2007 Psychological Factors affecting medical conditions in consultation-liaison psychiatry. In Porcelli P. Sonino N. (eds): Psychological factors affecting Medical Conditions. A New Classification for DSM V. Adv Psychosom Med. Basel, Krager, 2007, vol 28, pp 127-140; https://pubmed.ncbi.nlm.nih.gov/26402717/ ; https://pubmed.ncbi.nlm.nih.gov/23122485/ ; https://pubmed.ncbi.nlm.nih.gov/27744422/

Tab. 2. 

DCPR (+) N=23 DCPR (-) N=18 t-test/ χ2 P-values 

Age (years) 71.0 (7.1) 70.7 (7.8) t=0.15 0.87

Gender (F/M) 5/18 2/16 χ2=0.37 0.32

Left HF (N, %) 13/23 (56.5) 11/18 (61.1) χ2=0.87 0.54

Right HF (N, %) 2/23 (8.6) 0/18 (0.0) χ2=0.21 0.33

Left and Right HF (N, %) 8/23 (34.7) 7/18 (38.8) χ2=0.85 0.54

HFrEF; EF < 50% N (%) 17/23 (73.9) 16/18 (88.8) χ2=0.69 0.43

HFpEF; EF > 50% N (%) 5/23 (21.7) 2/18 (11.1) χ2=0.44 0.37 

LVEF (mean ± SD) 41.4 (11.1) 41.3 (8.9) t=0.03 0.97

BMI (kg/m²) 26.7 (2.3) 25.0 (4.4) t=1.61 0.11

BDI-II total 16.0 (5.4) 11.0 (8.6) t=2.26 0.02 

BDI-II somatic 8.3 (3.7) 6.0 (4.1) t=1.94 0.06 

BDI-II cognitive/affective 7.7 (3.8) 5.0 (5.1) t=1.95 0.06 

PHQ-15 12.0 (2.9) 5.5 (2.3) t=7.68 < 0.001 

PSI stress 1.7 (2.1) 1.4 (1.7) t=0.54 0.59 

PSI psychological distress 14.5 (10) 5.3 (4.7) t=3.52 0.001 

PSI abnormal illness 

 behavior 0.3 (0.7) 0.2 (0.7) t=0.52 0.60 

PSI well-being 3.8 (2.2) 3.5 (2.6) t=0.35 0.72 

PSI quality of life 1.4 (1.0) 1.6 (1.4) t=-0.48 0.63 

Smoking status 18/5 15/3 χ2=0.02 0.54

Alcohol intake 12/11 12/6 χ2=0.23 0.41

Statins use 14/9 15/3 χ2=0.42 0.34

Aspirin use 7/16 6/12 χ2=0.02 0.56

Beta blockers 21/2 17/1 χ2=0.01 0.56

IL-6 (log, mean ± SD) 0.72 (0.30) 0.43 (0.34) t=2.87 0.006

TNF-alpha (log, mean ± SD) 1.86 (0.94) 2.46 (1.1) t=-1.84 0.07 

DCPR = Diagnostic Criteria for Research in Psychosomatics ; psychosomatic syndromes were reported as dichotomous variables : (e.g., DCPR (+) / DCPR (-)) ; HFrEF= Heart Failure with Reduced Ejection Fraction; HFpEF= Heart Failure with Preserved Ejection Fraction; LVEF= left ventricular ejection fraction ; BDI-II = Beck Depression Inventory-II ; PHQ-15 = Patient Health Questionnaire-15 ; PSI =Psychosocial Index : all these scores were reported as continuous variables ; smoking status was reported as dychotomus variables (never-past smokers/current smokers) ; alcohol intake reported as dychotomus variables (users/no-users). 

5. How are continuous variables expressed (mean ± SD, median (IQR), or else)? I could not find that neither in the statistics description.

“Results for demographic and clinical variables are in the Table 1. Continuous variables are expressed as mean ± SD.”

6. There is no mention what were the levels of cytokines. The values of cytokines should be added in Table 1 (measured values, not logarithmic ones, so it would be easier to understand if it is normal or elevated). There is written in the discussion section that "increased inflammatory activity" (line 274), but the proof is missing in the results section.

“To permit comparison with results from other studies, the cytokine data are presented as the mean±SD of the untransformed data (Tab.1). “

We have now eliminated the term “increased inflammatory activity ” for a better readability of the manuscript. “Our study demonstrated an association between IL-6 levels and psychosomatic syndromes while there was no such association with depression. These findings reflect the study by Jansky et al. [42] which found a significant relation between IL-6 levels and psychosomatic psychopathology (i.e., vital exhaustion), but no depression, in patients with coronary heart disease.”

7. It is unclear what the authors mean by "comorbidity" in table 1? I presume that the conditions mentioned in lines 187-190, even the case numbers do not match (n=29 in the table and n=27 in the text). It needs clarification in the table. In addition, what does it mean systemic disorders (line 102)?

Thank you for pointing this out, we have now corrected this error. We have duly corrected it in the text (N=29, 70.7%) as was correctly reported in the table 1. Most patients (N=29, 70.7%) had at least one co-occurring medical condition, mostly arterial hypertension (80.4%), dyslipidaemia (78.0%), diabetes (48.7%), chronic renal failure (31.7) and chronic obstructive pulmonary disease (26.8%). At least one DCPR syndrome was found in 23 (56.1 %) patients.

8. Some of the results are presented as text (lines 219-225). It would be easier to follow them if they were at the table.

We now presented the data in the table 1

9. I think that it would be easier to read if the article would be divided into subsections (e.g., methods: inclusion/exclusion criteria; cytokines evaluation; evaluation of psychosomatic and psychological factors, etc.), at least in separate paragraphs. In addition, I would recommend extending the psychosomatic evaluation part and describing in more detail what was evaluated, how it was done, and the ranges of scores of each questionnaire.

We have now created subsections according to the reviewer’s suggestion and included in the Methods the following sentences: “The DCPR is administered during a face-to-face interview and takes about 15-30 min to be completed. It includes items scored in a yes/no response format evaluating the presence of 12 psychosomatic syndromes: a patient may be positive to more than one syndrome at the same time.”

“The Psychosocial Index (PSI) was used for assessing for stress and psychosocial dimensions (stress, well-being, psychological distress, abnormal illness behaviour and quality of life) [55]. This self-rating questionnaire includes 55 items. Some questions involve specific responses, most require a yes/no answer, while others are rated on a Likert scale (0–3, from ‘not at all’ to ‘a great deal’). The following domains are covered: a) Stress: this section (items 13–20 and 22–30) is an integration of both perceived and objective stress, life events and chronic stress. It consists of 17 questions with a total score ranging from 0 to 17; b) Well-being: this section (items 31–36) covers different areas of well-being (i.e., positive relations with others (items 31, 32), environmental mastery (items 33, 34) and autonomy (items 35, 36)), with a score ranging from 0 to 6; c) Psychological distress: this section (items 37–51) consists of a checklist of symptoms addressing sleep disturbances, somatization, anxiety, depression and irritability. The total score may range from 0 to 45; d) Abnormal illness behavior: it allows the assessment of hypochondriacal beliefs and bodily preoccupations (items 52–54). The total score may range from 0 to 9; e) Quality of life (item 55): a simple direct question on quality of life is included. The score ranges from 0 to 4.”

10. I think that other statistical methods such as ROC analysis could also fit this study (just an idea for authors).

Thank you for this suggestion which we will take into consideration.

We are extremely grateful to the Reviewer for her/his constructive comments, allowing us to improve the quality of the manuscript, overall.

---

## [Decision Letter · Decision Letter 1]

28 Feb 2022

Psychosomatic syndromes are associated with IL-6 pro-inflammatory cytokine in heart failure patients.

PONE-D-21-35033R1

Dear Dr. Altamura,

We’re pleased to inform you that your manuscript has been judged scientifically suitable for publication and will be formally accepted for publication once it meets all outstanding technical requirements.

Kind regards,

Kenji Hashimoto, PhD

Section Editor

PLOS ONE

Additional Editor Comments (optional):

Reviewers' comments:

Reviewer's Responses to Questions

**Comments to the Author**

1. If the authors have adequately addressed your comments raised in a previous round of review and you feel that this manuscript is now acceptable for publication, you may indicate that here to bypass the “Comments to the Author” section, enter your conflict of interest statement in the “Confidential to Editor” section, and submit your "Accept" recommendation.

Reviewer #1: All comments have been addressed

2. Is the manuscript technically sound, and do the data support the conclusions?

Reviewer #1: Yes

3. Has the statistical analysis been performed appropriately and rigorously? 

Reviewer #1: Yes

4. Have the authors made all data underlying the findings in their manuscript fully available?

Reviewer #1: Yes

5. Is the manuscript presented in an intelligible fashion and written in standard English?

Reviewer #1: Yes

6. Review Comments to the Author

Reviewer #1: The Authors have made the requested changes and satisfactorly answered to the raised criticism. I have no further comments.

7. PLOS authors have the option to publish the peer review history of their article (what does this mean?). If published, this will include your full peer review and any attached files.

Reviewer #1: **Yes: **Yan Wei

---

## [Editor Report · Acceptance letter]

2 Mar 2022

PONE-D-21-35033R1 

Psychosomatic syndromes are associated with IL-6 pro-inflammatory cytokine in heart failure patients. 

Dear Dr. Altamura:

I'm pleased to inform you that your manuscript has been deemed suitable for publication in PLOS ONE. Congratulations! Your manuscript is now with our production department. 

Kind regards, 

on behalf of

Prof. Kenji Hashimoto 

Section Editor

PLOS ONE